Excessive alcohol consumption and binge drinking in college students

Herrero-Montes Manuel 1 2
Alonso-Blanco Cristina 3
Paz-Zulueta María maria.paz@unican.es 1 4
Pellico-López Amada 1 5
Ruiz-Azcona Laura 1
Sarabia-Cobo Carmen 1 2
Boixadera-Planas Ester 6
Parás-Bravo Paula 1 2
1 Departamento de Enfermería, Universidad de Cantabria , Santander , Spain
2 Research Nursing Group, IDIVAL , Santander , Spain
3 Department of Physiotherapy, Occupational Therapy, Rehabilitation, and Physical Medicine, Universidad Rey Juan Carlos , Móstoles , Spain
4 GI Derecho Sanitario y Bioética, GRIDES, IDIVAL , Santander , Spain
5 Cantabria Health Service , Suances , Spain
6 Universitat Autónoma de Barcelona , Barcelona , Spain
Prazeres Filipe
Electronic publication date: 2022 May 4
Publication date: 2022
Volume: 10
Electronic Location ID: e13368
Received 2021 Dec 20; Accepted 2022 Apr 11
Copyright: ©2022 Herrero-Montes et al.
Copyright year: 2022
Copyright holder: Herrero-Montes et al.
License: This is an open access article distributed under the terms of the Creative Commons Attribution License, which permits unrestricted use, distribution, reproduction and adaptation in any medium and for any purpose provided that it is properly attributed. For attribution, the original author(s), title, publication source (PeerJ) and either DOI or URL of the article must be cited.
License URL: https://creativecommons.org/licenses/by/4.0/

Keywords: Binge drinking, Alcohol drinking in college, Alcohol-related disorders, Alcohol Use Disorders Identification Test

Funding: The University of Cantabria by means of the “Resolución de 4 de octubre de 2019 (R.R. 701/19), del vicerrector de Ordenación Académica y Profesorado, por la que se establecen las bases reguladoras y se convocan ayudas para contratos predoctorales de formación de profesorado universitario” This research was funded by the University of Cantabria by means of the “Resolución de 4 de octubre de 2019 (R.R. 701/19), del vicerrector de Ordenación Académica y Profesorado, por la que se establecen las bases reguladoras y se convocan ayudas para contratos predoctorales de formación de profesorado universitario”. The funders had no role in study design, data collection and analysis, decision to publish, or preparation of the manuscript.

==============================
Background

Binge drinking (BD) refers to a pattern of alcohol consumption characterized by the consumption of large amounts of alcohol in a short period of time followed by periods of abstinence. This drinking pattern is prevalent worldwide, mainly among young people. Excessive alcohol consumption is the spectrum of consumption patterns that may have or have had health consequences, and includes the concepts of risky alcohol use, harmful alcohol use and alcohol dependence according to Diagnostic and Statistical Manual of Mental Disorders, fourth edition (DSM-IV), the latter two are currently grouped into alcohol use disorder (AUD) according to the fifth edition of the DSM (DSM-5). Due to the high prevalence of BD among young people, especially university students, as well as the important consequences of its practice, a study was conducted to evaluate excessive alcohol consumption and its relationship with the practice of BD in university students.

Methods

A cross-sectional study was conducted among students (aged 18–30 years) enrolled in the academic year 2018–2019 at the Faculty of Nursing at a university in northern Spain. Data collection included sociodemographic information, and alcohol use information, collected using a semi-structured questionnaire. To measure the excessive alcohol consumption, this study used the Alcohol Use Disorders Identification Test (AUDIT).

Results

A total of 142 participants were included, of which 88.03% were women. Up to 38.03% were classified as BD. Up to 14.77% of non-BD participants and 66.67% of BD participants were classified as risky drinkers (AUDIT Total geq 8 in men or geq 6 in women) (p < 0.001). Up to 3.41% of the non-BD and 24.07% of the BD were drinkers with harmful alcohol use and probable alcohol dependence (AUDIT Total geq 13) (p < 0.001). A total of 5.68% of non-BD and 42.59% of BD were AUD drinkers (AUDIT Total geq 9 in males or geq 8 in females) (p < 0.001). In addition, statistically significant differences were found between the BD and non-BD groups in the responses to each of the AUDIT items, as well as in the total score and also in the scores of the three domains of the questionnaire.

Conclusions

Excessive alcohol consumption is frequent among university students, especially among those who practice BD.

Introduction

Binge drinking (BD) refers to a pattern of alcohol consumption characterized by the consumption of large amounts of alcohol in a short period of time followed by periods of abstinence. Several attempts have proposed definitions of BD, but the possible criteria are still under debate. The amount of alcohol ingested is a common parameter, although some also consider gender, speed of drinking, frequency or other parameters (Maurage et al., 2020). In general, alcohol affects women more intensely due to their lower levels of dehydrogenase enzymes, which are responsible for breaking down alcohol, and their higher body fat and water ratio. This causes alcohol levels to rise more rapidly after ingestion in women than in men. Consequently, women are more vulnerable to the harmful effects of alcohol (Milic et al., 2018). Thus, the National Institute on Alcohol Abuse and Alcoholism (2004) defined BD as “a drinking pattern that brings the amount of alcohol in the blood to 0.08g/dl (0.08%) or more. This typically occurs after a woman consumes 4 alcoholic drinks or a man consumes 5 alcoholic drinks within 2 hours”. In Spain, Parada et al. (2011) in an attempt to capture all the parameters that have been shown to be relevant in the definition, proposed that BD is the “consumption of 6 or more alcoholic drinks for men (60 grams), 5 or more for women (50 grams) on a single occasion (within a 2-hour period) at least once in the past 30 days”.

It is a pattern of consumption that is found worldwide, mainly among young people, reaching its peak between the ages of 20–24 years. The prevalence, in those over 15 years of age, in 2016, was 18.2%, with the European region having the highest rates, at 26.4% (World Health Organization, 2018). Among 15–16 year olds in Europe, The European School Survey Project on Alcohol and Other Drugs 2019, indicated that more than half of the students had consumed alcohol at least once in their lifetime; 47% reported having consumed alcohol in the past 30 days and 13% of students reported having been drunk in the past 30 days (ESPAD Group, 2020).

Excessive alcohol consumption is the spectrum of consumption patterns that may have or have had health consequences and which includes the terms: risky alcohol use, harmful alcohol use and alcohol dependence, the latter two are currently grouped into alcohol use disorder (AUD) (Curry et al., 2018).

Risky alcohol use corresponds to the pattern of substance use that increases the risk of harmful consequences for the user. Depending on the authors, it is used to refer to increased physical, mental and social risk. This last dimension would include risk of harm to third parties (World Health Organization, 1994).

Harmful alcohol use is understood as the pattern of consumption that causes harm. The harm may be physical or mental. Although this pattern often causes social problems as well, its diagnosis is not justified by this aspect alone (World Health Organization, 1994). This term appears in the International Classification of Diseases, tenth edition (ICD-10) and in the Diagnostic and Statistical Manual of Mental Disorders, fourth edition (DSM-IV).

Alcohol dependence refers to the existence of the need to consume alcoholic beverages on a regular basis in order to feel well or not to feel bad (World Health Organization, 1994). This is gathered in the ICD-10 and is also reflected in the DSM-IV.

In the year 2013, the American Psychological Association (APA) published the fifth edition of the DSM (DSM-5) which includes some important changes in the terminology of alcohol-related disorders. The terms harmful drinking and dependence were merged and replaced by AUD. This pathology is characterized by physical and behavioral symptoms including abstinence, tolerance, and intense craving. The subject continues to consume despite suffering from problems derived from such consumption patterns (American Psychiatric Association, 2013).

Globally, alcohol consumption was the seventh leading risk factor for both deaths and DALYs in 2016 (GBD 2016 Alcohol Collaborators, 2018). In addition, there are relationships between the average volume of consumption and more than 60 diseases, with the pattern of consumption modulating this relationship (Rehm et al., 2004).

Harmful alcohol use caused about three million deaths worldwide in 2016, mostly in men, accounting for 5.3% of all deaths in that period. This increase in mortality is especially high in the age group between 20 and 39 years, where 13.5% of deaths are attributed to alcohol (World Health Organization, 2018). In addition, the harmful alcohol use has an effect on the risk of reproductive pathology, cancer, cardiovascular diseases and diabetes, digestive pathology, infectious diseases, neuropsychiatric disorders, injuries and accidents, among others (World Health Organization, 2018).

BD often results in alcohol intoxication; this state causes impairments in cognitive abilities, including decision-making and impulse control, and impairments in motor skills, such as balance and hand-eye coordination, increasing the risk of death, injury, accidents, risky sexual behavior, sexual assault, and other harms, in addition to the direct consequences of intoxication, such as hangovers, blackouts, memory loss, nausea, and vomiting (White & Hingson, 2013; Kuntsche et al., 2017).

Alcohol consumption with a BD pattern in college students is implicated in a percentage of suicide attempts, memory loss, vandalism, and legal problems in young people (White & Hingson, 2013).

In the long term, despite numerous findings that binge drinking at an early age is associated with late consequences, such as AUD or other psychological disturbances, it is still unclear whether these long-term consequences are the effects of BD per se, or whether they are reflective of: regular heavy drinking; a broader “problematic behavior syndrome” associated with other social problems; or comorbidities such as antisocial personality disorder and delinquency, where early onset of this practice is sometimes only one of many symptoms, and not the primary or sole cause of the long-term consequences (Kuntsche et al., 2017).

Due to the high prevalence of BD among young people, especially university students, as well as the important consequences of its practice, a study was conducted to evaluate excessive alcohol consumption and its relationship with the practice of BD in university students.

Materials & Methods

Hypothesis

Excessive alcohol consumption is a frequent practice among college students, with particular intensity among those with a BD pattern.

Study design

A cross-sectional study was performed. The study design was approved by the Ethics Committee of Cantabria, Spain (Code: 2015.102). All procedures were conducted according to the Declaration of Helsinki (World Medical Association, 2013) and participants read and signed a written consent form prior to their participation in the study. The data were anonymized and treated confidentially in accordance with the Personal Data Protection legislation (Spanish Government Bulletin, 2018).

Participants

All students enrolled in the 2018–2019 academic year in a Faculty of Nursing at a university in northern Spain aged between 18 and 30 years were included in the study. Of 303 students enrolled, 142 finally participated in the study (Fig. 1).

Variables and measurement instruments

A semi-structured interview was conducted in which sociodemographic variables were collected: gender, age, place of residence and parents’ level of studies.

Binge drinking

The question How many standard drinks have you consumed at the most in a two-hour period in the last 30 days? Was used to classify participants as belonging to the BD or non-BD group. Concretely, to be considered BD, a participant had to answer that he/she consumed at least 5 standard drinks in the case of women and 6 standard drinks in the case of men according to the definition proposed by Parada et al. (2011).

Figure 1 Study participation diagram.

Excessive alcohol consumption

The Alcohol Use Disorders Identification Test (AUDIT) developed by the WHO was used to measure the concepts included in excessive alcohol consumption (Babor et al., 2001) which has been validated in Spain by Rubio Valladolid et al. (1998), and which has adequate psychometric properties in the detection of alcohol problems in college students (García Carretero et al., 2016). The authors have permission to use this instrument from the copyright holders.

The AUDIT is a 10-question questionnaire on recent consumption, symptoms of dependence and alcohol-related problems, with a total score that ranges from 0 to 40 points. The first eight questions are scored from 0 to 4 and questions nine and ten are scored at 0, 2 or 4 points.

This questionnaire is further divided into three domains: questions 1 to 3 refer to risk drinking, questions 4 to 6 to symptoms of dependence and questions 7 to 10 to harmful drinking (Babor et al., 2001).

The cut-off points for detecting risk consumption in the university population in Spain are set at eight points for men and six points for women. Furthermore, 13 points is the cut-off point for harmful consumption and probable alcohol dependence in both genders (García Carretero et al., 2016).

In relation to the AUD according to the DSM-5, after an exploration of pertinent literature, there are no studies that propose a score of the AUDIT questionnaire in university students in Spain to detect this disorder. However, Hagman (2016) proposes, in U.S. college students, nine or more items in males and eight or more items in females to identify college students with AUD criteria according to the DSM-5.

The form of administration can be verbal or self-administered. Verbal administration is recommended because of the possibility it offers to clarify ambiguous answers in patients with reading problems or to provoke immediate feedback and be able to start with brief advice in clinical contexts. The self-administered form requires less time and can provide more precise answers. The self-administered form was chosen for the study. The estimated completion time is 2–5 min.

Procedure

Participants were recruited through informative sessions and posters displayed at the faculty. Two researchers oversaw data collection. Two independent offices with good lighting and temperature conditions were available for this task. Data collection was carried out between December 2018 and January 2020.

Statistical analysis

For the statistical analysis, a distinction was made between categorical and quantitative variables. A descriptive bivariate analysis of BD and the remaining study variables was performed. Categorical variables are presented as counts and percentages of the total and of the BD and non-BD groups. These descriptive results have been completed with Pearson’s Chi-Square test to contrast the independence between each variable and BD. Quantitative variables are presented with basic descriptive statistics (n, mean, median, SD and quartiles) for the total and for the BD and non-BD groups. These results are complemented with the non-parametric Kruskall Wallis test to contrast the distributions of the quantitative variable between non-BD and BD. In cases of normal distribution of the variable in each non-BD and BD group, the results of the t-test are presented to compare the means of the independent groups: non-BD and BD.

Statistical analysis was performed using SAS v9.4 software, SAS Institute Inc, Cary, NC, USA. Statistical decisions were made with a significance level of 0.05.

Results

In total, 142 participants were included in the study. Eighty-eight participants belonged to the non-BD group and 54 to the BD group, corresponding to 61.97% and 38.03% of the total sample, respectively. Eleven participants (7.75% of the total) were abstainers, meaning that they had not consumed alcohol during the last year and 6 of these had never consumed alcohol (4.22% of the total). Women constituted 88.03% of the participants (Table 1).

Table 1 Socio-demographic variables.

	Total (n = 142)	Binge drinkers	
					No (n = 88)	Yes (n = 54)	
	n	%	n	%	n	%	
Gender												
Female	125	88.03%	81	92.05%	44	81.48%	
Male	17	11.97%	7	7.95%	10	18.52%	
				p = 0.059b	
	Mean	SD	Range	Median	Mean	SD	Range	Median	Mean	SD	Range	Median	
Age	20.73	2.67	18–30	20	20.77	2.77	18–30	20	20.67	2.53	18–28	20	
					p = 0.9839a	
Place of residence													
Family home	113	79.58%	68	77.27%	45	83.33%	
Not in the family home	29	20.42%	20	22.73%	9	16.67%	
					p = 0.3849b	
Maternal level of studies													
University	43	30.28%	33	37.5%	10	18.52%	
Secondary/vocational training	57	40.14%	32	36.36%	25	46.3%	
Primary	36	25.35%	21	23.86%	15	27.78%	
No studies	6	4.23%	2	2.27%	4	7.41%	
					p = 0.0639b	
Paternal level of studies													
University	45	31.69%	27	30.68%	18	33.33%	
Secondary/vocational training	55	38.78%	36	40.91%	19	35.19%	
Primary	36	25.35%	22	25,00%	14	25.93%	
No studies	6	4.23%	3	3.41%	3	5.56%	
					p = 0.8679a	
Notes.

a Wilcoxon Test.

b Chi-Square Test; SD (standard deviation).

Excessive alcohol consumption

In relation to the responses to each of the AUDIT items (Table 2):

Table 2 AUDIT Questionnaire items.

	Total (n = 142)	Binge drinkers	
			No (n = 88)	Yes (n = 54)	
	n	%	n	%	n	%	
1. How often do you have a drink containing alcohol?	
Never	14	9.86%	14	15.91%	0	0.00%	
Monthly or less	45	31.69%	40	45.45%	5	9.26%	
2 to 4 times a month	61	42.96%	29	32.95%	32	59.26%	
2 to 3 times a week	22	15.49%	5	5.68%	17	31.48%	
4 or more times a week	–	–	–	–	–	–	
		p < 0.001a	
2. How many standard drinks containing alcohol do you have on a typical day when drinking?	
1 to 2	73	51.41%	59	67.05%	14	25.93%	
3 to 4	46	32.39%	21	23.86%	25	46.30%	
5 to 6	15	10.56%	6	6.82%	9	16.67%	
7 to 9	4	2.82%	1	1.14%	3	5.56%	
10 or more	4	2.82%	1.	1.14%	3	5.56%	
		p <  0.001a	
3. How often do you have six or more drinks on one occasion?	
Never	83	58.45%	67	76.14%	16	29.63%	
Less than monthly	37	26.06%	18	20.45%	19	35.19%	
Monthly	16	11.27%	3	3.41%	13	24.07%	
Weekly	6	4.23%	0	0.00%	6	11.11%	
Daily or almost daily	–	–	–	–	–	–	
		p <  0.001a	
4. During the past year, how often have you found that you were not able to stop drinking once you had started?	
Never	122	85.92%	83	94.32%	39	72.22%	
Less than monthly	16	11.27%	5	5.68%	11	20.37%	
Monthly	4	2.82%	0	0.00%	4	7.41%	
		p <  0.001a	
5. During the past year, how often have you failed to do what was normally expected of you because of drinking?	
Never	123	86.62%	83	94.32%	40	74.07%	
Less than monthly	16	11.27%	4	4.55%	12	22.22%	
Monthly	3	2.11%	1	1.14%	2	3.70%	
		p  <  0.001a	
6. During the past year, how often have you needed a drink in the morning to get yourself going after a heavy drinking session?	
Never	117	82.39%	80	90.91%	37	68.52%	
Less than monthly	17	11.97%	6	6.82%	11	20.37%	
Monthly	5	3.52%	2	2.27%	3	5.56%	
Weekly	3	2.11%	0	0	3	5.56%	
		p = 0.003a	
7. During the past year, how often have you had a feeling of guilt or remorse after drinking?	
Never	90	63.38%	67	76.14%	23	42.59%	
Less than monthly	45	31.69%	18	20.45%	27	50.00%	
Monthly	6	4.23%	2	2.27%	4	7.41%	
Weekly	1	0.70%	1	1.14%	0	0.00%	
		p <  0.001a	
8. During the past year, how often have you been unable to remember what happened the night before because you had been drinking?	
Never	96	67.61%	73	82.95%	23	42.59%	
Less than monthly	39	27.46%	14	15.91%	25	46.30%	
Monthly	7	4.93%	1	1.14%	6	11.11%	
		p <  0.001a	
9. Have you or someone else been injured as a result of your drinking?	
No	123	86.62%	82	93.18%	41	75.93%	
Yes, but not in the last year	13	9.15%	5	5.68%	8	14.81%	
Yes, during the last year	6	4.23%	1	1.14%	5	9.26%	
		p = 0.009a	
10. Has a relative or friend, doctor or other health worker been concerned about your drinking or suggested you cut down?	
No	130	91.55%	86	97.73%	44	81.48%	
Yes, but not in the last year	6	4.23%	1	1.14%	5	9.26%	
Yes, during the last year	6	4.23	1	1.14%	5	9.26%	
		p = 0.003a	
Notes.

a Chi-Square Test; AUDIT (Alcohol Use Disorders Identification Test).

For the 1st question: How often do you drink alcoholic beverages? The percentage of people who indicated “2 or 3 times a week” and “2 to 4 times a month” is higher in BD participants (p < 0.001).

For the 2nd question: How many alcoholic beverages do you usually consume in a normal drinking day? The percentage of people who indicated “1 or 2” is higher among non-BD participants (p < 0.001).

For the 3rd question: How often do you drink six or more alcoholic beverages in a single day? The percentage of people who indicated “Never” was higher among non-BD participants (p < 0.001).

For the 4th question: How often in the course of the last year have you found that you could not stop drinking once you had started? The percentage of people who indicated “Less than once a month” and “Monthly” was higher in BD participants (p = 0.004).

For the 5th question: How often in the last year have you not been able to do the activity you were supposed to do because of drinking? The percentage of people who responded “Never” was almost 95% for non-BD participants and almost 75% for BD participants (p = 0.002).

For the 6th question: How often in the course of the last year have you needed to drink on an empty stomach in the morning to recover from heavy drinking the night before? The percentage of people who indicated “Never” was higher in non-BD participants (p = 0.003).

Regarding the 7th question: How often in the course of the last year have you had remorse or feelings of guilt after drinking? The percentage of people who indicated “Never” was higher in the non-BD participants (p < 0.001).

For the 8th question: How often in the course of the last year have you been unable to remember what happened the night before because you had been drinking? The percentage of people who indicated “Never” was higher in the non-BD participants (p < 0.001).

For the 9th question: Have you or someone else hurt yourself as a result of your drinking? The percentage of people who indicated “Never” was higher in the non-DB participants than in the non-DB participants (p = 0.009).

For the 10th question: Has a family member, friend, doctor, or health professional ever been concerned about your drinking or suggested that you stop drinking? The percentage of people who indicated “Never” was higher in non-BD participants (p = 0.003).

The median total direct score on the AUDIT questionnaire in non-BDs was 2 [Q1 = 1, Q3 = 4] points and 7 [Q1 = 5, Q3 = 12] points in BD (p < 0.001). In AUDIT domain 1, which refers to hazardous alcohol use and is composed of questions 1 to 3 of the questionnaire, the median score for Non-BDs was 2 [Q1 =1, Q3 =3] points and for DBs it was 4 points [Q1 = 3, Q3 = 6] (p < 0.001). In domain 2 of the AUDIT which refers to dependency symptoms and is composed of questions 4 to 6 of the questionnaire, the median score of the Non-BDs is 0 [Q1 = 0, Q3 = 0] points (more than 75% of the non-BDs score 0) whereas for the BDs it was 1 [Q1 = 0, Q3 = 2] (p < 0.001). In domain 3 of the AUDIT, which refers to Harmful alcohol use and is composed of questions 7 to 10 of the questionnaire, the median score of the non-BD is 0 [Q1 = 0, Q3 = 1] points and that of the “BD” is 2 [Q1 = 1, Q3 = 4] points (p < 0.001) (Table 3).

Table 3 AUDIT. Score and domains.

	Total (n = 142)	Binge drinkers	
					No (n = 88)		Si (n = 54)	
	Mean	SD	Range	Median	Mean	SD	Range	Median		Mean	SD	Range	Median	
Total AUDIT score	4.96	4.55	0–20	3	2.94	2.92	0–14	2		8.26	4.83	1–20	7	
					p <  0.001a	
	n	%	n	%		n	%	
Risky drinker (Total AUDIT score ≥ 8 in men or ≥ 6 in women)								
No	93	65.49%	75	83.23%		18	33.33%	
Yes	49	34.51%	13	14.77%		36	66.67%	
					p <  0.001b	
Drinker with harmful alcohol use y probable alcohol dependence (Total AUDIT score ≥ 13)								
No	126	88.73%	85	96.59%		41	75.93%	
Yes	16	11.27%	3	3.41%		13	24.07%	
					p <  0.001b	
Drinker with AUD (Total AUDIT score ≥ 9 in males or ≥ 8 in females)										
No	114	80.28%	83	94.32%		31	57.41%	
Yes	28	19.72%	5	5.68%		23	42.59%	
					p <  0.001b	
AUDIT DOMAINS	
Domain 1 Hazardous alcohol use	2.99	2.11	0–9	3	2.01	1.5	0–6	2		4.57	2	1–9	4	
					p <  0.001a	
Domain 2 Dependence symptoms	0.58	1.15	0–7	0	0.24	0.76	0–5	0		1.13	1.44	0–7	1	
					p <  0.001a	
Domain 3 Harmful alcohol use	1.4	2.14	0–11	1	0.69	1.34	0–8	0		2.56	2.65	0–11	2	
					p <  0.001a	
Notes.

a Wilcoxon test.

b Chi-Square Test AUDIT (Alcohol Use Disorders Identification Test); SD (Standard deviation); AUD (Alcohol use disorder).

Up to 14.77% of Non-BD participants and 66.67% of BD participants were classified as risky drinkers (AUDIT Total>= 8 in men or >= 6 in women) (p < 0.001), with a higher proportion of risky drinkers in BD participants. Up to 3.41% of the Non-BD and 13 participants (24.07%) of the BD were drinkers with harmful alcohol use and probable alcohol dependence (AUDIT Total >= 13) (p < 0.001), with a greater proportion of drinkers with harmful alcohol use and probable alcohol dependence in BD participants. A total of 5.68% of non-BD and 42.59% of BD were AUD drinkers (AUDIT Total>= 9 in males or >= 8 in females). There were statistically significant differences (p < 0.001) in the distribution of the variable according to BD, with a higher proportion of drinkers with AUD in BD participants (Table 3).

Discussion

Excessive alcohol consumption is the spectrum of consumption patterns that may have or have had health consequences and which includes the terms: risky alcohol use, harmful alcohol use and alcohol dependence according to DSM-IV criteria, the latter two are currently grouped into alcohol use disorder (AUD) according to DSM-5 (Curry et al., 2018).

Regarding risky alcohol use, the percentage of students in the BD group who were risky drinkers was higher than those who were not BD. Risky drinking, understood as the pattern of consumption that increases the probability of suffering physical, psychological and/or social consequences, is carried out by almost 35% of the entire sample: 67% of the BD and only 15% of the non-BD. Data from the EDADES 2017 survey in Spain shows a much lower prevalence, around 5% for the general population and 7% for young people between 15 and 24 years of age (Observatorio Español de las Drogas y las Adicciones, 2019). It should be noted that the EDADES Survey uses the AUDIT total score ≥8 points as the cut-off point for risky consumption, and in our study we used an AUDIT ≥8 for men and ≥6 in women according to the criteria proposed by García Carretero et al. (2016) for the university population in Spain; thus, it is to be expected that the number of risky drinkers would increase in a study such as ours, in which the sample is mainly female. For this reason, alcohol consumption in women in our sample may be higher. Some authors defend that the cut-off points or the amount of alcohol drinks ingested should be different between men and women (National Institute of Alcohol Abuse and Alcoholism, 2004; Parada et al., 2011; García Carretero et al., 2016; Hagman, 2016; Milic et al., 2018; Maurage et al., 2020) because in general, alcohol affects women more intensely due to their lower levels of dehydrogenase enzymes, which are responsible for breaking down alcohol, and their higher body fat and water ratio (Milic et al., 2018).

García Carretero et al. (2016) in university students, found a prevalence also lower than that of our study, 20.1%, however, in his case he excluded those with an AUDIT score ≥13 points from the group of risky drinkers (García Carretero et al., 2016). Given the relationship between BD and risky alcohol consumption it would be interesting to implement programs to reduce consumption in all students and specifically among those who practice BD, brief alcohol interventions have proven useful in preventing harm from consumption in college students (Hennessy et al., 2019). However, the educations programs of prevention approaches have been found ineffective when conducted independent. The motivational interventions (building motivation to change drinking, challenging expectations about alcohol’s effects, correcting misperceptions through normative feedback, providing cognitive-behavioral skills training) are more effective but can be very costly (Mastroleo & Logan, 2014). Perhaps it would be interesting to include a gender perspective due to the differences in consumption between men and women.

The approach is very complex and some authors propose new intervention approaches to approach the university population using the new technology like short message service, web applications and mobile apps. Currently, the evidence on their efficacy is limited, but they are fundamental lines of research for the prevention of alcohol consumption in young people in the future (Mastroleo & Logan, 2014; Staiger et al., 2020).

Regarding harmful alcohol use and probable alcohol dependence, approximately 25% young people who are BD suffer negative consequences from their drinking pattern and have a possible alcohol dependence, this percentage is reduced to 3% non-BD young people. The total prevalence is almost double that reported by García Carretero et al. (2016) for a university population in Spain, which was 6.4%. The difference may be due to the fact that our sample is mostly female and already other works have found higher prevalence of BD in female university students (Bartoli et al., 2014). The detrimental effects of alcohol consumption occur mainly through three interrelated mechanisms: the toxicity of alcohol on various organs, the development of alcohol dependence that promotes loss of self-control and is often associated with mental illness and social problems, and the psychoactive effects of alcohol intoxication (Babor et al., 2010). The BD drinking pattern often results in alcohol intoxication; this state causes impairments in cognitive abilities, including decision-making and impulse control, and impairments in motor skills, such as balance and hand-eye coordination, increasing the risk of death, injury, accidents, risky sexual behavior, sexual assault, and other harms, in addition to the direct consequences of intoxication, such as hangovers, blackouts, memory loss, nausea, and vomiting. (Kuntsche et al., 2017; White & Hingson, 2013). In addition, BD pattern drinking in college students is implicated in a percentage of suicide attempts, memory loss, vandalism, and legal problems in young people (White & Hingson, 2013). It seems clear that, if we were to reduce BD-type drinking, we would achieve a reduction in the harmful effects of alcohol consumption that seem to be especially related to this type of consumption in young people.

Our study also analyzed the AUDIT total score with the scores proposed by Hagman (2016) (AUDIT females ≥8 and AUDIT males ≥9) to detect AUD according to DSM-5 criteria in college students in the USA, finding that this disorder affects approximately 40% of BD and only 6% of non-BD, in contrast to the reference study that obtained a 61.75% prevalence among BD, although the context and definition of BD are different. After an exploration of pertinent literature, our work is the first study in our country to analyze the relationship of BD with AUD using these AUDIT scores, therefore, the results should be taken with caution.

The scores of all the individual items and their grouping into AUDIT domains show statistically significant differences between the BD and non-BD groups, i.e., considering the three domains, BD subjects drink more frequently and in greater quantities; they present more symptoms of alcohol dependence; and they have suffered more harmful consequences of consumption than non-BD subjects. Although BD is a practice that involves a large amount of alcohol consumption in a short period of time but that is carried out on a non-continuous basis, usually weekly, it produces consequences not only because of people become intoxicated, but also because symptoms of dependence are detected in those who practice it.

This study presents data on the responses to each of the AUDIT items and also compares the scores for each item between BD and non-BD students. It is unusual to find such specific information in scientific studies and we consider that it may be useful to gain a greater understanding both of the impact of alcohol consumption, in general, and the practice of BD in particular.

Regarding the limitations, it should be noted that there is no universal definition of BD, and therefore the results of the different studies are difficult to compare. However, to minimize this, the study used the definition by Parada et al. (2011) which is adapted to the Spanish population.

Another limitation is that the sample was composed exclusively of nursing students, mostly women, which makes it difficult to generalize the results obtained. In future research, it would be interesting to develop a definition of BD that can be adapted to the different circumstances of each country, to enable comparison of the results obtained by the different studies.

In addition, only the presence of depression was evaluated with the Beck Depression Inventory, not finding any relevant results in the sample studied. The presence of possible others psychiatric conditions was not assessed to control for the possible effects of these conditions on the relationship between BD and excessive alcohol consumption.

Conclusions

Excessive alcohol consumption is frequent among university students, especially among those who practice BD, and this has negative consequences on their health. It would be important to assess alcohol consumption in this population and implement measures focused on BD practice.

Supplemental Information

Supplemental Information 1 Raw data

The participants’ responses to each of the questions.

Click here for additional data file.

Supplemental Information 2 Raw data codebook

Click here for additional data file.

Supplemental Information 3 Test Instrument Permissions

Click here for additional data file.

We would like to thank the invaluable participation of all those who generously volunteered to collaborate in this study.

Additional Information and Declarations

Competing Interests

Author Contributions

Human Ethics

Data Availability

The authors declare there are no competing interests.

Manuel Herrero-Montes, María Paz-Zulueta and Paula Parás-Bravo conceived and designed the experiments, performed the experiments, analyzed the data, prepared figures and/or tables, authored or reviewed drafts of the paper, and approved the final draft.

Cristina Alonso-Blanco conceived and designed the experiments, analyzed the data, prepared figures and/or tables, authored or reviewed drafts of the paper, and approved the final draft.

Amada Pellico-López and Laura Ruiz-Azcona performed the experiments, authored or reviewed drafts of the paper, and approved the final draft.

Carmen Sarabia-Cobo conceived and designed the experiments, performed the experiments, analyzed the data, authored or reviewed drafts of the paper, and approved the final draft.

Ester Boixadera-Planas analyzed the data, authored or reviewed drafts of the paper, and approved the final draft.

The following information was supplied relating to ethical approvals (i.e., approving body and any reference numbers):

The Ethics Committee of Cantabria, Spain (protocol code: 2015.102).

The following information was supplied regarding data availability:

The raw data are available in the Supplementary File.

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
