# Peer review of "Excessive alcohol consumption and binge drinking in college students"

_PeerJ, doi:10.7717/peerj.13368_

## Round 0.1 · original submission · Major Revisions

The present manuscript needs to be substantially improved. The authors should submit the revised manuscript along with a point-by-point response to the reviewers' comments.

Reviewer 1 ·

Basic reporting

The work by Herrero-Montes and colleagues provides an interesting approach to the relationship between binge drinking and alcohol use disorders.

However, several points need to be clarified:

1.- Some important references have not been mentioned in the paper (or are not referred to in the latest version). For instance, the references to some of the epidemiological data could be updated to provide more accurate information (e.g., "ESPAD Report 2019: Results from the European School Survey Project on Alcohol and Other Drugs [Internet]. 2020."; or "Griswold, M. G., Fullman, N., Hawley, C., Arian, N., Zimsen, S. R., Tymeson, H. D., ... & Farioli, A. (2018). Alcohol use and burden for 195 countries and territories, 1990–2016: a systematic analysis for the Global Burden of Disease Study 2016. The Lancet, 392(10152), 1015-1035."). Also it seems that are affirmations that would have a better backup if they were accompanied by proper references (i.e., line 61 "depending on the authors"; lines 244-246)

2.- There are seemingly contradictory affirmations that need to be clarified. For example, "Binge drinking (BD) refers to a pattern of alcohol consumption for which, currently, there is no consensus on its definition or terminology" (lines 42,43). However, afterwards, the authors did not provide any references to this apparent lack of consensus; instead, they provided, in fact, two consensual definitions by the NIAAA (2004) and their adaptation to the Spanish population (Parada et al., 2011). If the authors are interested, they could check the recent paper by Maurage and colleagues (2020), where insightful contributions are made on this matter (Maurage, P., Lannoy, S., Mange, J., Grynberg, D., Beaunieux, H., Banovic, I., ... & Naassila, M. (2020). What we talk about when we talk about binge drinking: Towards an integrated conceptualization and evaluation. Alcohol and alcoholism, 55(5), 468-479.). In this same regard, the affirmation "BD often results in alcohol intoxication" (line 87) seems to be in contradiction with the definition provided where it is said that binge drinking is "a drinking pattern that brings the amount of alcohol in the blood to 0.08g/dl (0.08%) or more." (line 46, 47).

Also, the use of terms like "harmful alcohol use", "alcohol dependence", and "alcohol use disorders" should be reviewed to provide a more consistent argumentation. In this regard, it is not clear why the authors did not stick to the most recent definition (DMS-V and Curry et al., 2018) and why they refer to the previous versions of the DSM and ICD. This critical conceptualization issue can hamper the study's conclusions and should be carefully addressed. Indeed, the sentence that forms the manuscript's conclusions affirm that BD is associated with alcohol dependence AND alcohol use disorders (AUD), but taking into account the latest definition provided by the authors (DMS-V and Curry et al., 2018) these terms are partially overlapping.

Is it possible to talk about alcohol dependence and alcohol use disorders as independent entities in the author's opinion? The sentence

3.- It is really interesting to include the paragraph from lines 96 to 102. In which way do the authors think that their current investigation tries to overcome the questions pointed out by Kuntsche et al. (2017)? For example, do the authors have any way to measure different alcohol consumptions pattern regarding their sample? Have they tried to isolate binge drinking effects from those of more regular "heavy drinking"?
About this same set of affirmations by Kuntsche et al. (2017), do the authors assess the presence of potential psychiatric conditions to control the potential effects of these conditions on the relationship between binge drinking and "excessive alcohol consumption". If not, it should be stated as part of the study's limitations.

4.- Why the authors did not propose any hypothesis?

5.- The discussion would be more precise if the authors used the same unit of measurement to compare results (i.e., lines 261-265; 2 out of 10 proportion vs percentage).

6.- There is an erratum in the line 195 where it says non-DB it should say non-BD

Experimental design

no comment

Validity of the findings

Which was the criteria to classified as binge drinkers those participants who reported that their typical consumption (Q2 of AUDIT) was 1-2 drinks and/or that they never had more than 6 drinks (Q3)?

Are there significant differences in sex distribution within the whole sample?

This could be an important question as the previous literature have reported sex-related differences in the effects of binge drinking (Wilsnack, R. W., Wilsnack, S. C., Gmel, G., & Kantor, L. W. (2018). Gender differences in binge drinking: Prevalence, predictors, and consequences. Alcohol research: current reviews.) and so should be address in the discussion of the results.

·

Basic reporting

General comments
The article focuses on a major global problem and social vice: the trend of harmful alcohol consumption. The article attempts to define the subject and also offers a survey study on it. However, certain improvements should be strongly considered in lieu of a major revision.

Experimental design

Concept comprehension
One striking question in this entire study is “what is binge drinking?”. The question was not answered in the abstract and in the first few lines of the introduction (lines 42-44), but these parts rather created a more confusing and vaguer announcement of the concept. Line 42 states “…currently there is no consensus on its definition or terminology…” but in contradiction to this claim, offered a definition from an accredited body in the following sentence. These opposing views should be clarified. It is understandable that the word “consensus” was used, however, since there is an explanation for the term by the National Institute on Alcohol Abuse and Alcoholism, that can be used. It is suggestable for the authors to use the NIAAA definition in the abstract and opening segment of the introduction to offer the scientific audience, scholar, or stakeholder a fundamental understanding.

Understanding of technical scope
Some other important questions have been left unanswered. In the entirety of the manuscript, the alcohol consumption limit standard for men is higher than that for women. This is stated in lines 48-52. Even for data categorization, lines 148-149 state “The cut-off points for detecting risk consumption in the university population in Spain are set at 8 points for men and 6 points for women.” No technical basis for this classification was offered in the entire manuscript. Thoughts such as difference in hormonal biochemistry, muscular makeup, lifestyle activity, etc could come up out of curiosity in the comprehension of any audience, yet without any standard clarification. Recent literature should be explored to answer this question and then properly incorporated and cited in the manuscript.
In a similar course, no justification for the point system attributed to each answer to the questions in the questionnaire was given. For example, lines 149-150 states “Furthermore, 13 points is the cut-off point for harmful consumption and probable alcohol dependence in both genders” THE CURIOUS QUESTION IS WHY? Why not a lesser number such as 12, or a higher number such as 20? A concise but descriptive summary should be offered regarding the questionnaire arrangement. Citing the literature from which it was adapted might not suffice.
This should be done for the study variables too.

Validity of the findings

Inadequate discussion

The entire discussion section of this manuscript is a more formal declaration of results along with a comparison with other literature in scarce amounts. For each stated result, the cause/origin should be clearly discussed. Discussion regarding the factors for the continuous trend of such results over time should follow. Thereafter, optimal and workable solutions should be offered with citations from recent literature. Thoughts such as government regulations, societal views, medical opinions, physiological and psychological effects, and nutritional opinions should also be included to develop a befitting discussion section.

A simple example can be taken from lines 247-248 “Regarding risky alcohol use we found that the percentage of students in the BD group who were risky drinkers was higher than those who were not BD”. The subsequent line should declare what Regarding risky alcohol use means. Is it related to the body mass index (BMI)? Is it related to medical history? Should a hypertensive patient take the same amount of alcohol with a health patient? What are the standards of some notable governments (such as European Union) on the amount of alcohol that should be consumed, or the amount of alcohol that beverage producers can add to their beverages? Are there any formal requirements to declare the alcohol contents on the beverage labels/packages? Who is a student in this context? A teenager in the same school and an adult (above 40 years) in the same school and class – can the same consumption standards be recommended for them? This orientation should be used for reconstructing the discussion aspect.

Additional comments

Vague conclusion.

Authors should consider constructing a conclusion that crisply describes the problem studied, the result of the problem studied, and RECOMMENDATION(S) for the solution of the problem. The conclusion of the manuscript should be rewritten.

Technical writing

Abstract
The entire abstract section should be reconstructed. For example, the first three sentences are seemingly vague and sanding alone without many connections. The abstract should be an executive and technical summary of the whole study.

Personal pronouns
In too-many parts of the manuscript, the use of “we” was excessive. This should be removed and “the study” should be used as the sentence subject in such cases. “To the best of OUR knowledge…” can be corrected to “after an exploration of pertinent literature…” etc

---

## Round 0.2 · Minor Revisions

The authors have addressed most of the issues raised in the first round of review and now only minor corrections are suggested.

Reviewer 1 ·

Basic reporting

no comment

Experimental design

no comment

Validity of the findings

no comment

Additional comments

The authors have addressed all of the issues raised in the first review. These changes have significantly improved the quality of the manuscript.

I do not have any other comments

·

Basic reporting

The article has been revised and its quality has been significantly improved. However, certain minor corrections are suggested to further garnish its scholastic appeal.

Experimental design

The stated hypothesis seems like a tautological declaration.

The authors wrote in the background "Binge drinking (BD) refers to a pattern of alcohol consumption characterized by the CONSUMPTION OF LARGE AMOUNTS of alcohol in a short period of time followed by periods of abstinence. This drinking pattern is prevalent worldwide, mainly among young people."

The written hypothesis states "the hypothesis was that university students with a pattern of BD consumption had a higher prevalence of excessive alcohol consumption."

The manuscript has already explained that binge drinking is a type of excessive alcohol consumption. Having that as the hypthesis is analogous to stating an hypothesis stating that all boys are males.

The hypthesis might be in a fashion of proving that college students in particular are notabaly into the practice of binge drinking.

Validity of the findings

The authors stated from the abstract that the "...we conducted a study to evaluate excessive alcohol consumption and its relationship with the practice of BD in university students."

Nonetheless, the about 4-word line conclusion section is comprised of a vague repition of the BD definition and a declaration about prevention and early detection.

Considering the study's intent and justification, it is expectable that the conclusion should be a comprehensive summary of the finalized findings with regards to evaluation on alcohol consumption and the social practise of BD amongst early university scholars.

The conclusion should be restructured.

Additional comments

The authors stated from the abstract that the "...we conducted a study to evaluate excessive alcohol consumption and its relationship with the practice of BD in university students."

The use of personal pronouns in technical writings are not too ethical. Technical writings are meant to be as objective as they should be. Personal pronouns offers a compromise to this. A suggested approach is " ...the study was conducted to..."

This should be corrected in the entire manuscript.

---

## Round 0.3 · accepted · Accept

The authors have satisfactorily responded to all the questions made by the referees and made the necessary changes to the manuscript.